

**A model for interpreting the deformation mechanism of reservoir landslides in the**
**Three Gorges Reservoir area, China**
Zongxing Zou[1], Huiming Tang[1], Robert E. Criss[2], Xinli Hu[3], Chengren Xiong[1], Qiong
Wu[3], Yi Yuan[4]
[1]Three Gorges Research Center for geo-hazards, China University of Geosciences, Wuhan, 430074,
China
[2]Department of Earth and Planetary Sciences, Washington University, One Brookings Drive, Saint
Louis, United States
[3]Faculty of Engineering, China University of Geosciences, Wuhan, 430074, China
[4]Department of Land and Resources of Hubei Province, Wuhan, 430074, China
*Correspondence author*: Huiming Tang (tanghm@cug.edu.cn)


**Abstract.** Landslides whose slide surface is gentle near the toe and relatively steep in the middle
and rear part are common in the Three Gorges Reservoir area, China. The mass that overlies the
steep part of the slide surface is termed the "driving section" and that which overlies the gentle part
of the slide surface is termed the "locking section". A driving-locking model is presented to elucidate
the deformation mechanism of reservoir landslides of this type, as exemplified by Shuping landslide.
More than 13 years of field observations that include rainfall, reservoir level and deformation show
that the deformation velocity of Shuping landslide depends strongly on the reservoir level but only
slightly on rainfall. Seepage modelling shows that the landslide was destabilized shortly after the
reservoir was first impounded to 135 m, which initiated a period of steady deformation from 2003 to
2006 that was driven by buoyancy forces on the locking section. Cyclical water-level fluctuations in
subsequent years also affected slope stability, with annual "jumps" in displacement coinciding with
drawdown periods that produce outward seepage forces. In contrast, the inward seepage force that
results from rising reservoir levels stabilizes the slope, as indicated by decreased deformation
velocity. Corrective transfer of earth mass from the driving section to the locking section
successfully reduced the deformation of Shuping landslide, and is a feasible treatment for huge
reservoir landslides in similar geological settings.
**Keywords**: Three Gorges Reservoir, Reservoir landslide, Water level fluctuation, Deformation
mechanism, Shuping landslide





**1 Introduction**
Reservoir landslides attract wide attention as they can cause huge surge waves and other
disastrous consequences (Huang et al., 2017; Wen et al., 2017; Froude and Petley, 2018). The surge
wave produced by the 1963 Vajont landslide in Italy destroyed Longarone village and caused nearly
2,000 fatalities (Paronuzzi and Bolla, 2012). A similar surge associated with the 2003 Qianjiangping
landslide, which slipped shortly after the Three Gorges Reservoir (TGR) in China was first
impounded, capsized 22 fishing boats and took 24 lives (Xiao et al., 2007; Tang et al., 2019). To
ensure the safety of the reservoir, 1.5 billion US dollars have been invested to reinforce the reservoir
banks in TGR. However, reinforcement structures are costly and difficult to construct, and thus many
huge reservoir landslides have not been treated (Wang and Xu, 2013). Many remain in a state of
continuous deformation, such that cumulative monitored displacements of several meters are now
documented at the Huangtupo (Tang et al., 2015; Dumperth et al., 2016), Outang (Yin et al., 2016),
and Baishuihe (Li et al., 2010; Du et al., 2013) landslides. Additional study of the deformation and
failure mechanisms, and risk reduction strategies of these huge reservoir landslides is of great
significance.
Most research on the deformation or failure mechanism of reservoir landslides involves
numerical modelling, physical model testing, or field observation. Many numerical simulations have
studied how landslide geometry, material permeability, variation rate of water level and pressure
variation influence the stability of reservoir landslides (Rinaldi and Casagli, 1999; Lane and Griffiths,
2000; Liao et al., 2005; Cojean and Cai, 2011; Song et al., 2015). Both small-scale (Junfeng et al.,





2004; Hu et al., 2005; Miao et al., 2018) and large-scale physical model experiments (Jia et al., 2009)
have been conducted to investigate the deformation features of reservoir landslides related to water
level change. Casagli et al. (1999) and Rinaldi et al. (2004) monitored the pore water pressure in
riverbanks to determine its effect on bank stability.

Since the impoundment of TGR, monitoring systems have been installed on or within many

reservoir landslides (Ren et al., 2015; Huang et al., 2017; Song et al., 2018; Wu et al., 2019), which
provide valuable data for the study of their deformation features. Several studies show that reservoir
water level variations and rainfall are the most critical factors that govern the deformation velocities
of reservoir landslides in TGR (Li et al., 2010; Tang et al., 2015; Ma et al., 2016; Wang et al., 2014).
Unfortunately, the effects of rainfall and reservoir level are difficult to distinguish because the period
of TGR drawdown is managed to coincide with the rainy season. Detailed deformation studies that
incorporate long-term continuous monitoring data are needed to quantify how periodic water-level
variations affect reservoir landslides. Moreover, the evolutionary trend of these deforming landslides
and feasible treatments for these huge reservoir landslides are rarely studied.

This study presents a model combined with seepage simulations to elucidate how reservoir

landslides deform, using the Shuping landslide as an example. The new environmental and
deformation data provided here extend the observational period for this landslide to more than 13
years, and include results that confirm the effectiveness of a control strategy that have been
implemented.

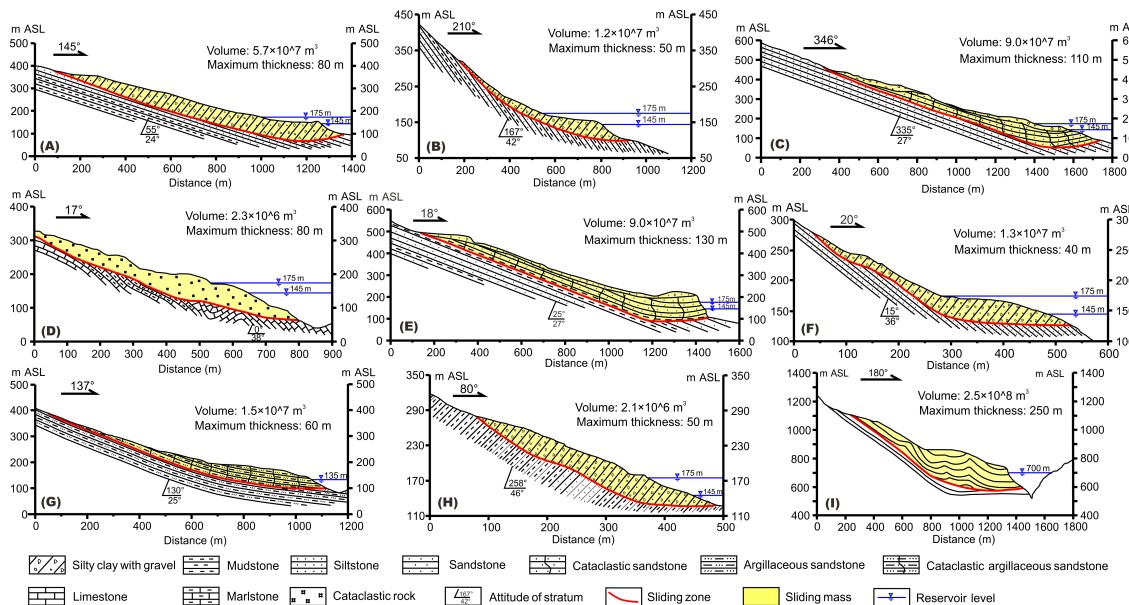

**Fig. 1** Geological profiles for typical reservoir landslides, all in the TGR except Vajont in Italy (**I**).

(**A**) Jiuxianping landslide (Wang, 2013); (**B**) Xicheng landslide (Song, 2011); (**C**) Outang landslide

(Yin et al., 2016); (**D**) No.1 riverside slump of Huangtupo landslide (Wang et al., 2014); (**E**)

Muyubao landslide (Lu, 2012); (**F**) Baishuihe landslide (Lu, 2012); (**G**) Qiangjiangping landslide

(Xiao et al., 2007); (**H**) Ganjuyuan landslide (Qin, 2011); (**I**) Vajont landslide, the world famous

reservoir-induced landslide in Italy (Paronuzzi and Bolla, 2012). See Fig. 2 for locations.

## 2 A geomechanical model for reservoir-induced landslide

### 2.1 Typical reservoir-induced landslides in the Three Gorges Reservoir

Figure 1 and Fig. 2 summarize the reservoir landslides of most concern in the TGR plus the world famous Vajont landslide. These landslides have many common features. First, all these landslides have large volumes, ranging from millions of cubic meters to tens of millions of cubic meters, and all are difficult to reinforce by conventional structures such anti-slide pile, retaining wall etc. Second, the front part of the slide mass is always thicker than the rear part, with a maximum thickness from 40 m to over 100 m. Another important feature of these profiles (Fig. 1) is that the slope of the slide surface decreases gradually from the rear to the front and may become horizontal or even anti-dip in the front. Last, these landslides were reactivated after the reservoir impoundment, with large observed deformations indicating their metastable situation. All these features are relevant to the deformation behavior of reservoir landslides, as discussed below.

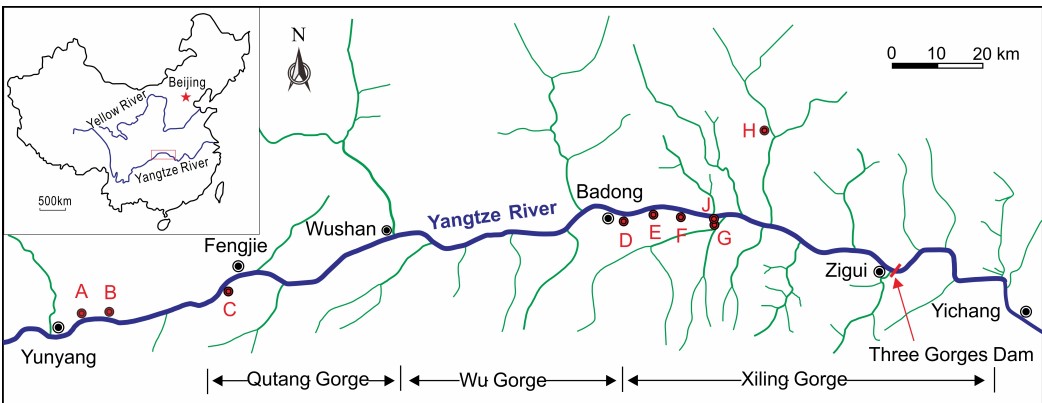

**Fig. 2** Location map for important landslides in TGR. Jiuxianping landslide (A); Xicheng landslide (B); Outang landslide (C); Huangtupo landslide (D); Muyubao landslide (E); Baishuihe landslide (F); Qiangjiangping landslide (G); Ganjuyuan landslide (H); Shuping landslide (J), Case study.




**2.2 Driving-locking model**
Due to the relatively high slope of the slide surface in the middle and rear part, the slide force
exceeds the resistance force on the proximal slide surface, producing extra thrust on the lower-front
slide mass. Consequently, the rear-upper is termed the "driving section" (Fig. 3). In contrast, the
potential slide surface underlying the lower-front part of the slide mass provides more resistance due
to the relatively gentle slide surface slope and greater thickness of the slide mass. The lower-front
part of the slide mass is termed the "locking section" (Fig. 3), as it blocks the driving section, thereby
playing a critical role in landslide stability (Tang et al., 2015).

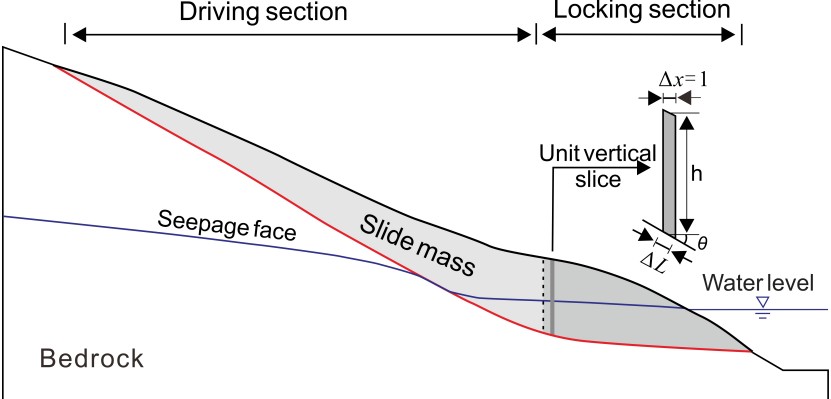


**Fig. 3** Driving-locking model for reservoir landslide
The locking section is defined as the lower-front part of the slide mass, where each unit vertical
slice (Fig. 3) can be self-stabilized under its self-weight. According to the limit equilibrium method
and the definition of the locking section, the sliding force of each vertical slice is the component of
its gravitational force along the slide surface, which cannot exceed the shear resistance provided by
the base. The special position where the sliding force of the vertical slice equals the resistance force



provided by the slide surface is regarded as the boundary between the driving and locking sections.
Force balance along the sliding direction for this special vertical slice can be written as
$$w \sin \theta_1 = w \cos \theta_1 \tan \varphi + c \Delta L \qquad (1)$$
where $w$ is the weight of the unit vertical slice; $\theta_1$ is the slope angle of the slide surface at the
boundary between the driving and locking sections; $\Delta L$ is the length of the slice base (see Fig. 3);
and $c$ and $\varphi$ are the cohesion and internal friction angle of the slide surface, respectively.

The weight of the slice $w=\gamma h \Delta x$, where $\gamma$ is the unit weight of the slide mass, $h$ is the vertical

distance from the center of the base of the slice to the ground surface, $\Delta x$ is the unit width of the slice,
and $\Delta L=\Delta x/\cos\theta_1$ (Fig. 3). Thus Eq. (1) can be rewritten as
$$\tan \theta_1 = f + k / \cos^2 \theta_1 \qquad (2)$$
where $f=\tan\varphi$, $k=c/\gamma h$.
The solution to Eq. (2) provides the slope angle $\theta_1$ of the slide surface:
$$\theta_1 = 0.5 \arcsin T \qquad (3)$$
where   $T = \dfrac{(2k + f) + \sqrt{(2k + f)^2 - 4k(k + f)(1 + f^2)}}{1 + f^2}$

Empirical values for the cohesion of the slide surface is less than 40 kPa, while the internal

friction angle of the slide surface varies between 10° and 25° (Chang et al., 2007), and the unit
weight of the soil is typically about 20 kN/m$^3$. In order to further elucidate the effect of various
parameters on the length of the locking section, contour maps of $\theta_1$ under different shear strength
parameters $c$ and $\varphi$ and the thickness of the slide mass $h$ are plotted (Fig. 4), as derived from Eq. (3).



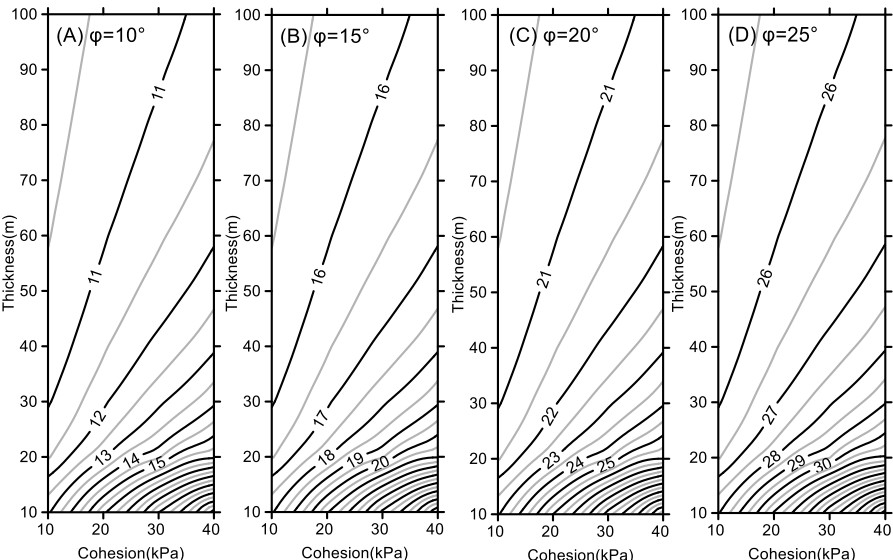


**Fig. 4** Coutour maps for the slope angle $\theta_1$ of slide surface that denotes the boundary between the

driving and locking sections under various shear strength parameters and slide mass thickness.
Figure 4 shows that $\theta_1$ increases as the internal friction angle $\varphi$ increases; however, by
comparison of the pattern and the values of the contour in the four sub-figures, the difference
between $\theta_1$ and $\varphi$ has little relationship to $\varphi$. Due to the effect of cohesion, $\theta_1$ is always larger than $\varphi$
as shown in Fig. 4. As the cohesion $c$ decreases, the difference between $\theta_1$ and $\varphi$ decreases, and for
cohesionless material with $c=0$, $\theta_1$ is equal to $\varphi$. Fig. 4 also shows that when the thickness of the slide
mass reaches about 40 m, the difference between $\theta_1$ and $\varphi$ is very small (less than 3°), which
becomes even less as the thickness increases. These results indicate that for the thick slide mass (up
to 40 m), the boundary between the locking and driving sections can be approximated as the position
where the slope angle $\theta_1$ equals the internal friction angle $\varphi$.



**2.3 Effect of water force on the locking and driving sections**
The impacts of the water level change on the reservoir slope stability can be quantified by
analyzing the changes in water force on the slope. Lambe and Whitman (2008) have demonstrated
that the water forces acting on an element of the slope can be equivalently expressed by either the
ambient pore-water pressure (Fig. 5A) or by seepage and buoyancy forces (Fig. 5B). The latter form,
i.e., seepage and buoyancy forces, are employed here to clarify the mechanical mechanism of water
force on the reservoir bank.

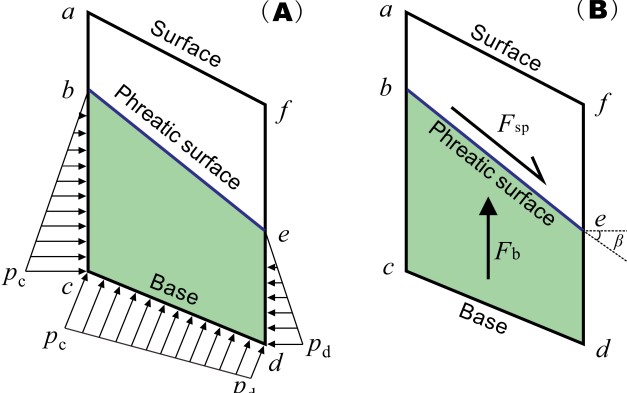


**Fig. 5** Two equivalent ways to display the water force acting on a slice of the slide mass. (A)
expressed by pore-water pressure; (B) expressed by the seepage force $F_{sp}$ and the buoyancy force $F_{b}$.
The seepage force ($F_{sp}$) represents the frictional drag of water flowing through voids that is
proportional to the hydraulic gradient and acts in the direction of flow. It can be expressed as (Lambe
and Whitman, 2008)

$$F_{sp} = \gamma_{w} i V \qquad (4)$$

Where $\gamma_{w}$ is the unit weight of water; $i$ is the hydraulic gradient and equals $\sin\beta$ where $\beta$ is the slope
angle of the phreatic surface; $V$ is the submerged volume of the analyzed element as the trapezoid





area enclosed by points *bcde* in Fig. 5.
When the groundwater flows outwards as occurs during reservoir level drops, the corresponding
outward seepage force decreases the slope stability. In contrast, the seepage force will be directed
inward during reservoir level rise, increasing slope stability.
The buoyancy force ($F_b$) of the water exerted on the element can be expressed as
$$F_b = \gamma_w V \tag{5}$$

The factor of safety (*Fos*) used to quantify the slope stability can be defined as the ratio of the
shear strength (resistance, $F_r$) along the potential failure surface to the sliding force ($F_s$) by the
Mohr-Coulomb failure criterion (Wang et al., 2014):
$$Fos = \frac{F_r}{F_s} = \frac{\sum_{j=1}^{n}\left[c\Delta L_j + N_j \tan\varphi\right]}{\sum_{j=1}^{n} w_j \sin\theta_j} \tag{6}$$

where $n$ is the total number of slices; $N$ is the normal force on the base of each slice, and the other
symbols are as above. Suppose that the variation of the effective slide mass weight in a slice is $\Delta w$,
due to the change of buoyancy force, which thereby modifies the resistance and sliding forces by $\Delta F_r$
and $\Delta F_s$ respectively. The corresponding change of the factor of safety $\Delta Fos$ is:
$$\Delta Fos = \frac{F_r + \Delta F_r}{F_s + \Delta F_s} - \frac{F_r}{F_s} = \frac{\Delta F_r * F_s}{\left(F_s + \Delta F_s\right) F_s}\left(1 - \frac{Fos}{\Delta F_r / \Delta F_s}\right) \tag{7}$$

The ratio of $\Delta F_r$ to $\Delta F_s$ for a vertical slice due to the change of its effective weight $\Delta w$ is
approximately:
$$\frac{\Delta F_r}{\Delta F_s} = \frac{\Delta w \cos\theta \tan\varphi}{\Delta w \sin\theta} = \frac{\tan\varphi}{\tan\theta} \tag{8}$$

Suppose that $\theta_2 = \arctan\left(\dfrac{\tan\varphi}{Fos}\right)$, where the change of the vertical slice weight has no influence





on the current stability ($\Delta Fos$=0). If $\theta<\theta_2$ and $\Delta w>0$, then $\Delta Fos>0$, indicating that increase of the
weight of lower-front part of the slide mass where its slope angle of the slide surface $\theta$ is less than $\theta_2$
will improve the stability of the whole slide mass; conversely, decrease of the weight of the
lower-front part would decrease stability. In contrast, the upper-rear part has a contrary tendency. As
mentioned above, continuously deformed reservoir landslides are metastable and their corresponding
*Fos* is around 1; hence $\theta_2 \approx \varphi$. Consequently, in the cases that reservoir landslide is under metastable
state and has a thickness up to 40 m, $\theta_1 \approx \theta_2 \approx \varphi$, the locking section and driving section have the same
mechanical behavior as described above. Either an increase in the weight of the locking section or a
decrease in the weight of the driving section will improve the stability of the slope and vice versa.

In summary, the effect of ground water on the slope or landslide stability can be resolved into a

seepage force and a buoyancy force. The effect of the seepage force on slope stability depends on the
direction and magnitude of flow. Buoyant forces change the effective weight of the slide mass and
have contrary effect on the locking and driving sections. On the basis of these rules, the mechanical
mechanism for reservoir-induced landslide can be illustrated as Fig. 6.


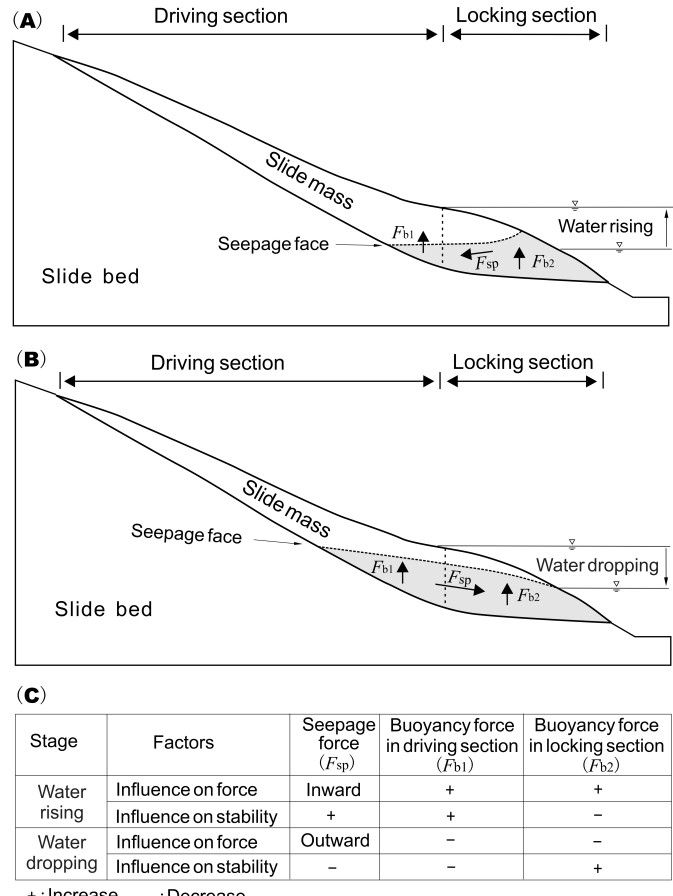


+ : Increase    − : Decrease

**Fig. 6** Mechanical mechanism for reservoir-induced landslide. (A) water level rise; (B) water level
drop; (C) effects of various mechanisms on the landslide stability during water level rise and drop.
**3 Shuping landslide**

Shuping landslide is located in Shazhenxi Town, Zigui County, Hubei Province, on the south

bank of the Yangtze River, 47 km upstream from the Three Gorges dam (Fig. 2). After the first
impoundment of the reservoir in 2003, serious deformation was observed that endangered 580
inhabitants and navigation on the Yangtze River (Wang et al., 2007). Previous studies of the Shuping
landslide utilized GPS extensometers (Wang et al., 2007), or field surveys (Lu et al., 2014) to clarify


the deformation. This study provides a detailed geomechanical model that includes seepage and
buoyancy effects to clarify the deformation mechanism of this landslide which is calibrated by
long-term monitoring data.

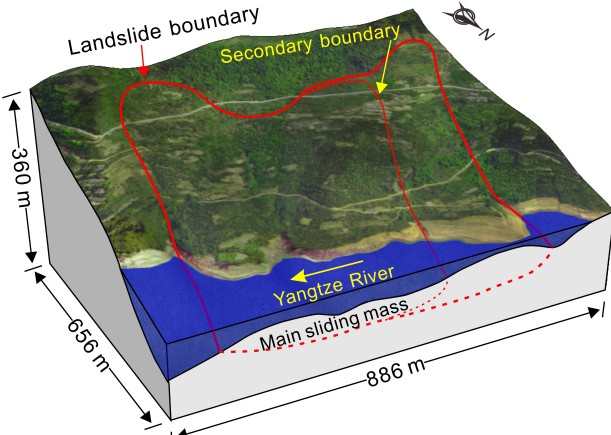


**Fig. 7** Full view of Shuping landslide (the surface satellite map © Google Maps).
**3.1 Geological setting**
The Shuping landslide is a chair-shaped slope that dips 20° to 30° to the north, toward the
Yangtze River (Fig. 7). The landslide is bounded on the east and west by two topographic gutters.
The altitude of its crown is 400 m above sea level (ASL), while its toe is about 70 m ASL, which is
now submerged by the reservoir, level of which varies annually between 145 and 175 m ASL (Fig. 8).
Borehole and inclinometer data (Lu et al. 2014) indicate that there are two major slide surface within
the west part of the slope and the upper rupture zone divides the slide mass into two parts (see Fig. 7).
The whole slide mass has a thickness of 30-70 m, a N-S length of about 800 m and W-E width of
approximately 700 m, constituting a total volume of ~27.5 million m$^3$, of which 15.8 million m$^3$
represents the main slide mass.
Shuping landslide is situated on an anti-dip bedrock of marlstone and pelitic siltstone of the
Triassic Badong Group (T₂b) (Fig. 9). The upper part of the slide mass is mainly composed of yellow
and brown silty clay with blocks and gravels, while the lower part of the slide mass mainly consists
of dense clay and silty clay with gravels, with a thickness of about 50 m on average. The deep
rupture zone is a 0.6~1.7 m layer that extends along the surface of bedrock, and consists of
yellowish-brown to steel gray silty clay. The upper rupture zone in the west part has similar
compostion and has an aveage thickness of 1.0-1.2 m. The dip angle of the slide surface decreases
gradually from the rear to the front (Fig. 9), so the driving-locking model is appropriate for Shuping
landslide. Before reservoir impoundment, boreholes ZK17 and ZK18 were dry but borehole ZK14
contained groundwater near the rupture zone.

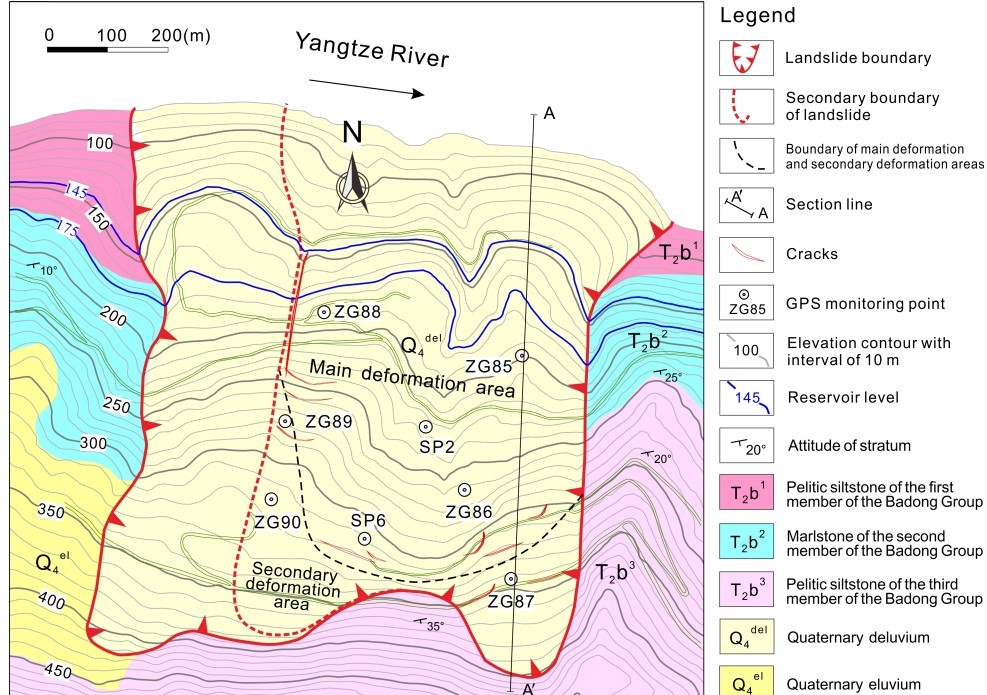

**Fig. 8** Engineering geology map of Shuping landslide

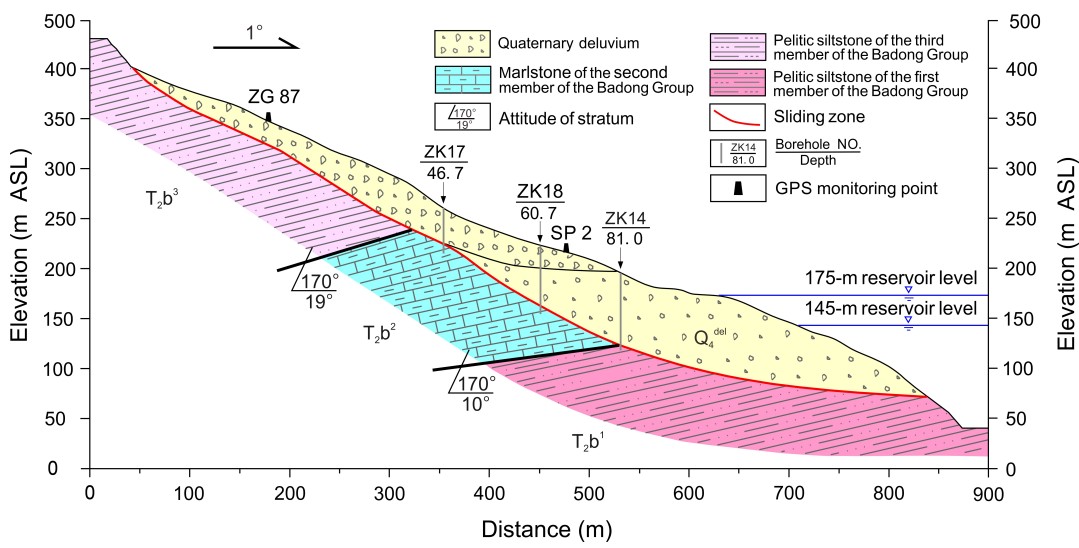


**Fig.9** Geological profiles along section A-A' as shown in Fig. 8
**3.2 Monitoring instrumentation**
The displacement monitoring system of Shuping landslide consists of 11 global positioning
system (GPS) survey points, three of which are datum marks that were installed on stable ground
outside the landslide area with the remainder being on the main slide mass (Fig. 8). Seven of the GPS
monitoring points (SP2, ZG85, ZG86, ZG87, ZG88, ZG89 and ZG90) were set in June 2003 and
GPS monitoring points SP6 was set in August 2007. All the GPS monitoring points were surveyed
every half month, and the system was upgraded to automatic, real-time monitoring in June 2012. The
daily rainfall records are obtained from the Meteorological Station near the Shuping landslide
(source: http://cdc.nmic.cn/). Daily reservoir level is measured by China Three Gorges Corporation
(source: http://www.ctg.com.cn/inc/sqsk.php).





## 3.3 Engineering activity

The evolution of Shuping landslide is related to four stages of human activity (Fig. 10). The first

stage was the 139 m ASL trial reservoir impoundment (from April 2003 to September 2006). The

reservoir water level was lifted from 69 to 135 m ASL and then changed between 135 and 139 m

ASL. The second stage was 156 m ASL trial reservoir impoundment (from September 2006 to

September 2008). The reservoir water level was raised from 139 to 156 m ASL, and then varied

annually between 145 and 156 m ASL. The third stage was 175 m ASL trial reservoir impoundment.

This stage began when the reservoir water level was raised to 175 m ASL, and thereafter managed to

annually varied between 145 and 175 m ASL (Tang et al., 2019). During the fourth stage, an

engineering project for controlling the deformation of Shuping landslide was conducted in

September 2014 and completed in June 2015.

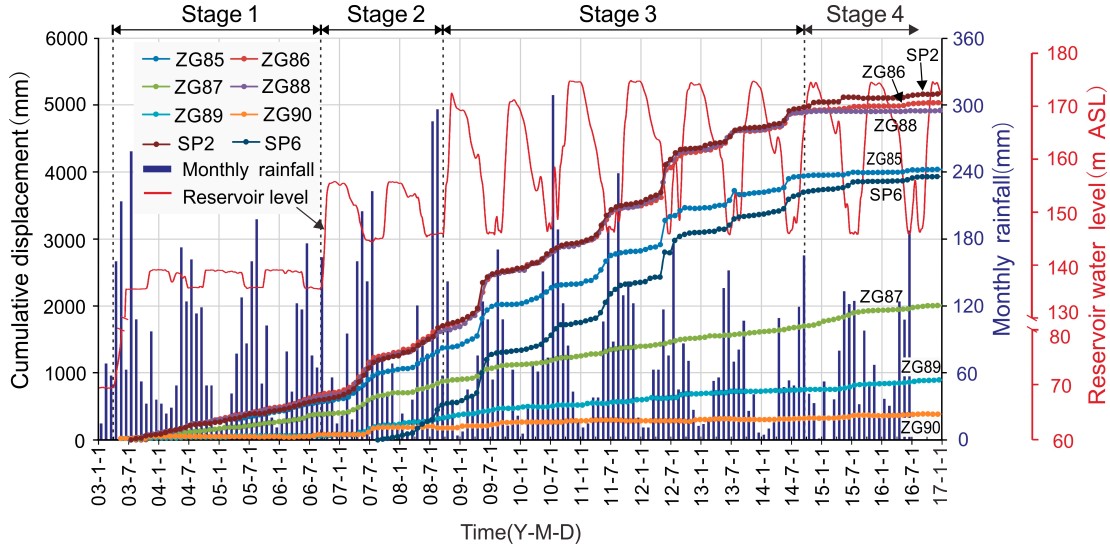

**Fig. 10** Monitoring data for Shuping landslide from 2003 to 2016.





## 4 Field observational results

### 4.1 Overall deformation feature

According to the deformation features revealed by the GPS monitoring system (Fig. 10, Fig. 11) and field investigations, the main slide mass can be divided into a main deformation area and a secondary deformation area (Fig. 8). The main deformation area underlies most of the area and has a cumulative displacement up to 4-5 m, as measured at sites ZG85, ZG86, ZG88, SP2 and SP6. During the 13-year monitoring period point SP2 underwent the largest cumulative displacement (5.168 m), followed by ZG86 and ZG88 which recorded 5.039 m and 4.919 m, respectively. Deformations were essentially synchronous at the monitoring sites as indicated by the similar shape of their cumulative displacement curves, which typically show steady rises in the first impoundment stage, step-like trends in the second and third impoundment stages, and flat trends after the engineering treatment. Deformations were smaller and steadier in the secondary deformation area, as indicated by gentle cumulative displacement curves at ZG89, ZG90, and ZG87, which recorded cumulative displacements of 0.5-2 m during 2003 to 2016.

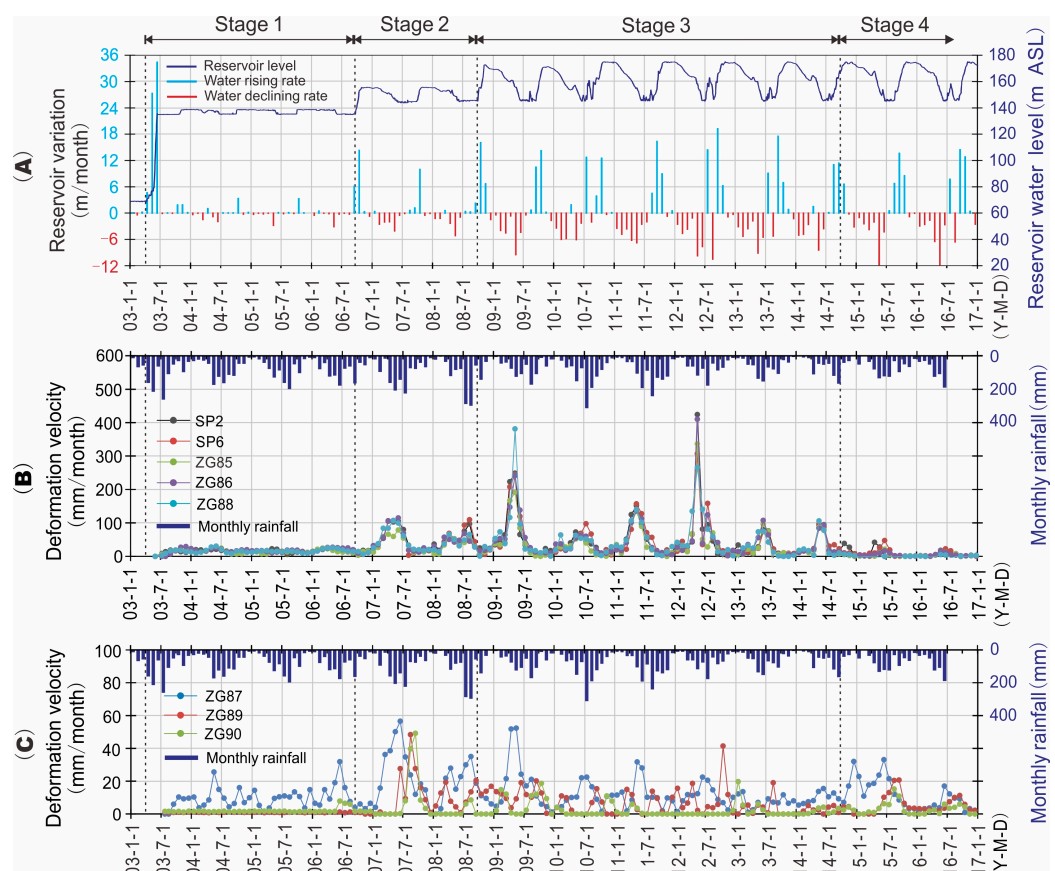

**Fig. 11** Time series of reservoir level, rainfall and landslide displacement from 2003 to 2016. (A)
Reservoir water levels and variation rates (positive for level rise, negative for level drop); (B)
Deformation velocity of the GPS points in the main deformation area and monthly rainfall; (C)
Deformation velocity of the GPS points in secondary deformation area and monthly rainfall.

**4.2 Deformation feature in different stages**

After the reservoir level first rose to 135 m ASL in June 2003, the main deformation area
deformed at an average velocity of 15.6 mm/month until September 2006, with each site recording
rather steady displacement curves whose tiny or nonexistent steps correspond to the small annual



variations in reservoir level. In contrast, no obvious deformation occurred during Stage 1 at ZG89
and ZG90 in the secondary deformation area.

During the earliest two months of Stage 2 (September, October 2006), when the reservoir level

first rose to 156 m ASL, deformation velocities of the main deformation area decreased to 13.4 and
9.7 mm/month respectively, indicating that slide mass stability had improved. For the next two
months (November, December) the velocity increased to 11.5 and 14.3 mm/month, as the reservoir
level was steady at 156 m ASL. During the subsequent drawdown period when the reservoir level
dropped to 145 m ASL in 2007, the deformation velocity increased to a maximum of about 100
mm/month (Fig. 11), resulting in an average "jump" of 458 mm in the cumulative displacement
curve, which then became flat while the reservoir remained at 145 m (Fig. 10).

During the beginning of Stage 3 when the reservoir first rose to nearly 175m in October 2008,

the deformation velocity of the main deformation area decreased to 12.7 mm/month, compared to 65,
74, 32 mm/month in the previous three months. Shortly after the reservoir rose to its highest level,
the level underwent a gradual decline and the deformation velocity increased steadily. The maximum
deformation velocity reached 378.6 mm/month at ZG88 in May 2009 when the water level declined
rapidly, a rate almost four times higher than when the reservoir dropped from 156 to 145 m ASL in
2007. Then the deformation velocity decreased to a relatively low value when the water level was
steady at 145 m ASL (Fig. 11B).

In the subsequent 6 years of Stage 3 the reservoir level underwent a series of similar annual

variations, and the slide mass responded with a series of deformation "jumps". During these cycles,





the deformation velocity decreased as the reservoir rose, maintained low values when the reservoir
remained high, began to increase as drawdown began, and attained the values up to 165 mm/month
when drawdown was rapid. The corresponding cumulative deformation curves featured obvious
"jumps" during drawdown periods, then became relatively flat as the reservoir was maintained at the
low level of 145 m ASL. Clearly, these results show that deformation velocity is high during
reservoir drawdown and low during reservoir rise.
After the engineering treatment was completed in June 2015, the "jumps" in the cumulative
displacement curves disappeared and the curves became very flat (Fig. 10). The deformation was
reduced to a low level of 4.1 mm/month in the main deformation area, demonstrating effective
treatment.

**4.3 Effect of water-level fluctuation and rainfall on the deformation of Shuping landslide**

The largest "jump" in the cumulative displacement curves averaged 479 mm and occurred in
May to June, 2012, while the second was the jump of 458 mm in May to June, 2009. These periods
corresponded with the two highest drawdown rates of 9.67 and 9.38 m/month, respectively (Fig.
11A). During these two years, rainfall amounts were relatively low with monthly maxima of 180
mm/month in 2009 and 190 mm/month in 2012 (Fig. 11). These data clearly demonstrate that the
deformation of Shuping landslide is primarily driven by reservoir level variations and not by rainfall.
This relationship is also confirmed by the low deformation velocities and flat cumulative
displacement curves during the July and August peak of the rainy season, when the reservoir is held
at its lowest level.

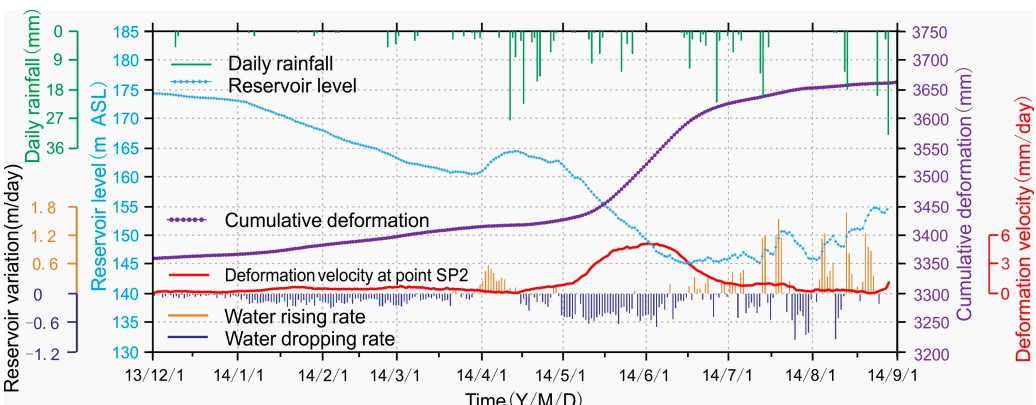


**Fig. 12** Monitoring data of GPS point SP2 on the middle part of slide mass, from December 2013 to
September 2014.


Figure 12 clarifies the influence of reservoir level and rainfall on landslide deformation. In
December 2013, the reservoir level dropped at an average rate of 0.041 m/day, and the corresponding
deformation velocity was 0.22 mm/day. In the subsequent three months, the drawdown rate of the
reservoir level increased to 0.147 m/day, and the deformation velocity rose to 0.54 mm/day. During
March 2014, the deformation velocity decreased as the water level increased, even though intense
rainfalls were recorded during this period (up to 27.5 mm/day). In the following rapid drawdown
period (0.419 m/day) from May to June, the deformation velocity increased to about 5 mm/day.
Subsequently, the deformation velocity decreased to less than 1.2 mm/day as the water level
remained low, although rainfall was abundant. These details confirm that the deformation velocity of
the Shuping landslide is positively related to the drop rate of the reservoir, with rainfall having little
effect.
Unlike the flat displacement curves and low deformation velocity in other years when the
reservoir level was steady at the lowest annual level in July and August, deformation velocities were
large in 2008 and 2010 (65.0 and 73.8 mm/month in July and August 2008; 58.4 mm/month in July





2010, about half of the average highest monthly deformation velocity, 165 mm/month, during rapid
draw down period). Very heavy rainfall was recorded during those periods, up to 300 mm/month.
However, August 2011 had the next heaviest rainfall of 250 mm/month, yet the cumulative
displacement curve remained flat and the deformation velocity was low (22.2 mm/month). These
data illustrate that heavy rainfall can decrease landslide stability and accelerate deformation, but
nevertheless is a secondary factor. The difference in the displacement velocity between the months
with the highest (2008, 2010) and the second highest (2011) levels of rainfall suggests that a
threshold exists, with rainfall exceeding this value having a significant effect but with less having
little significance. This threshold appears to be about 250-300 mm/month.
**5 Numerical simulation**

In this section, groundwater flow in the Shuping slope under the variation of the reservoir level

is simulated to assist the driving-locking model to explain the deformation process of Shuping
landslide. Seepage simulation is performed by the SEEP/W module of GEOSTUDIO software (see
http://www.geoslope.com). The deformation state of the landslide is usually regarded as the
performance of the landslide stability state (Wang et al., 2014; Huang et al., 2017). Thus, the *Fos*
(Safety of factor) of the Shuping landslide is calculated with the simulated groundwater level, to
evaluate the stability of the Shuping landslide under various impoundment scenarios. In this study,
the *Fos* of the Shuping landslide is calculated by Morgenstern-Price method (Zhu et al., 2005) using
the SLOPE/W module of GEOSTUDIO software. Different evaluation method for landslide stability
will lead to different value of *Fos*; thus we only employ the calculated values of *Fos* to investigate





the variation trend of the landslide stability.

Figure 13 shows the numerical simulation model of the Shuping landslide, whose framework is

based on the geological profile map in Fig. 9. The slope was divided into six regions composed of
five materials with different properties (Table 1). Zero flux boundary conditions were assigned along
the bottom horizontal and the right vertical boundaries. A constant water head was applied at the left
vertical boundary assuming that it is sufficiently far from the reservoir to not be affected by
reservoir-level variations. A series of inverse modelling tests and water tables at the boreholes were
adopted to determine the constant water head at the left vertical boundary. The optimum water head
at the left boundary is 230 m ASL. The hydrograph of TGR from January 1, 2003 to September 10,
2014 (Fig. 14(A)) and generalized hydrograph of the trial impoundment at 175 m ASL (Fig. 14(B))
were used to define the right boundary adjacent to the reservoir. Initial conditions were defined using
the water tables revealed by boreholes.

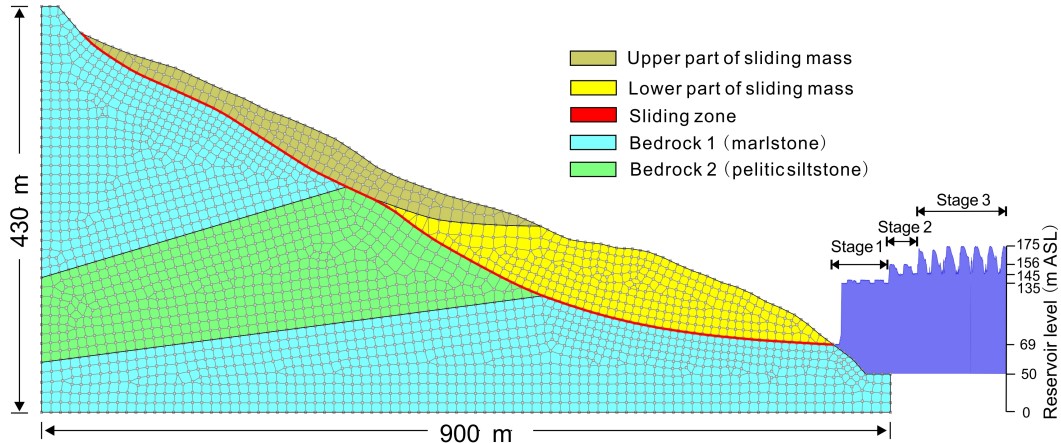

**Fig. 13.** Numerical simulation model of seepage for Shuping landslide.


**Table 1** Hydrologic and mechanical properties of Shuping landslide

| Location | Material | Saturated conductivity $k_s$(m/day) | Unit weight $\gamma$(kN/m³) | cohesion $c'$(kPa) | friction angle $\varphi'$ (°) |
|---|---|---|---|---|---|
| Upper part of slide mass | Silty clay with blocks and gravels | 4.95[a] | 20.3[a] | / | / |
| Lower part of slide mass | Silty clay with gravels | 3.90[a] | 20.3[a] | / | / |
| Rupture zone | Silty clay | $2.98*10^{-2}$[b] | / | 25.7[a] | 20.4[a] |
| Bedrock 1 | Marlstone | $1.47*10^{-4}$[b] | / | / | / |
| Bedrock 2 | Pelitic siltstone | $8.99*10^{-5}$[b] | / | / | / |

[a] Provided by Hubei Province Geological Environment Terminus (2003)
[b] Values of similar material from literature (Hu et al., 2015)

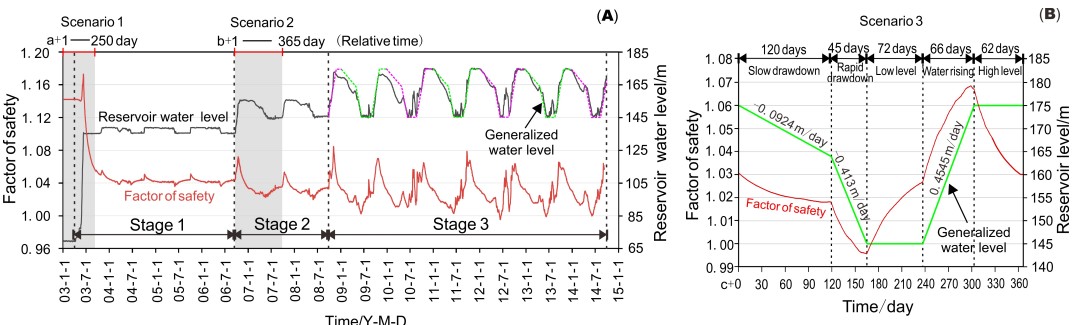


**Fig. 14** (A) Time series of reservoir level and corresponding calculated *Fos* of Shuping landslide
from January 1, 2003 to September 10, 2014. (B) Generalized annual variation curve of the reservoir
level obtained by fitting the real water level from 2008 to 2014 (Stage 3) and the corresponding time
series of the calculated *Fos* of Shuping landslide.
**5.1 Scenario 1: first trial impoundment at 139 m *ASL***

From April 10 to June 11, 2003 (a+100~162 day), the reservoir level rose rapidly from 69 to 135



m ASL. Fig. 15 shows that, during this period, groundwater storage increased in the toe of the slide
mass and within the lower part of the locking section, increasing buoyancy forces that destabilized
the slope. In contrast, the inwardly-directed flow created a seepage force directed towards the slope,
increasing stability. Owing to the high hydraulic gradient, the stabilizing effect of the seepage force
on the slope prevails over the destabilization due to increased buoyancy, so slope stability was
improved during this phase, as indicated by the increase in *Fos* up to 1.17 (Fig. 14).

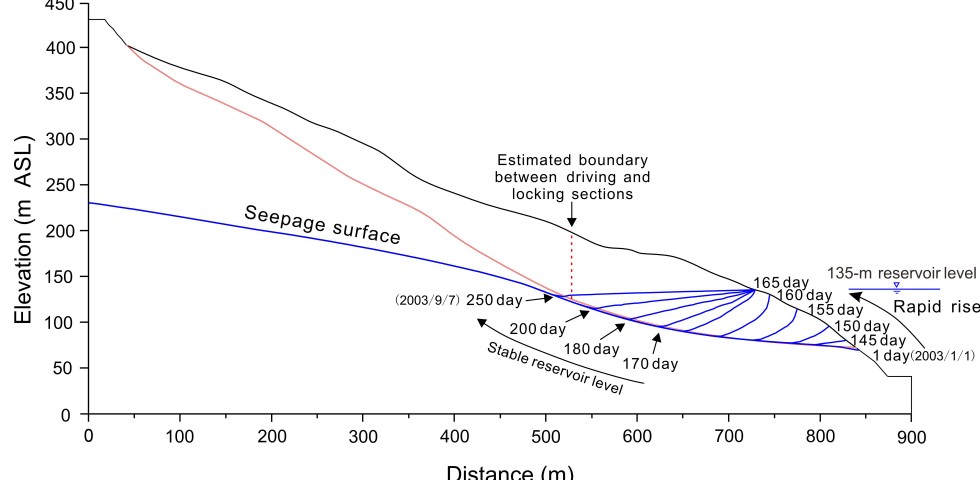


**Fig. 15** Simulated groundwater tables during the period of rapid reservoir rise from January 1, 2003
to September 7, 2003.

In the following period (a+163 day~), the reservoir level was maintained around 135 m ASL.

The water table progressively rose until it approximated the reservoir level. During this period, the
slope of the water table front decreased gradually, leading to a decrease of the seepage force in the
slope. At the same time, the buoyancy uplift effect increased steadily in the locking section as the
groundwater table rose (Fig. 15). The combination of a decreased seepage force and the increased
buoyancy led to a decrease in slope stability during this phase, so the *Fos* dropped below its initial



value of 1.142. Afterwards, the slope stability continued to decrease until the new but temporary state
of equilibrium was reached. The safety factor was around 1.045 as the reservoir level was maintained
around 135 m ASL.

The delay between the reservoir impoundment and the decrease in stability is consistent with the

creation of obvious cracks after the reservoir rose to 135 m ASL (Wang et al., 2007). The famous
Qianjiangping landslide (Fig. 2), which is located near the Shuping landslide and has similar
geological setting, occurred one month (13 July 2003) after the reservoir first rose to 135 m ASL
(Xiao et al., 2007).
**5.2 Scenario 2: first trial impoundment at 156 m *ASL***

During the periods when the water level rose from 135 m ASL to 156 m ASL (b+1~30 day) (Fig.

16), and stayed stable at 156 m ASL (b+30~138 day), the effects of ground water level change on the
stability of Shuping landslide were similar to the effects in scenario 1. When the reservoir level
dropped from 156 to 145 m ASL during the drawdown period of February to June (b+138~260 day),
groundwater flow towards the reservoir, thus creating an outward, destabilizing seepage force on the
slope. The computed factor of safety decreased gradually from 1.070 to 1.025, in agreement with the
observed increase in deformation velocity during this period. As the reservoir level was then
maintained at 145 m ASL (b+260~365 day), the transient seepage gradually transitioned to
steady-state seepage, accompanied by a progressively decline of the water table in the inside part of
the fluctuation zone, a weakening of the destabilizing effect of the seepage force, and a result of
increase in slope stability (*Fos*=1.035).

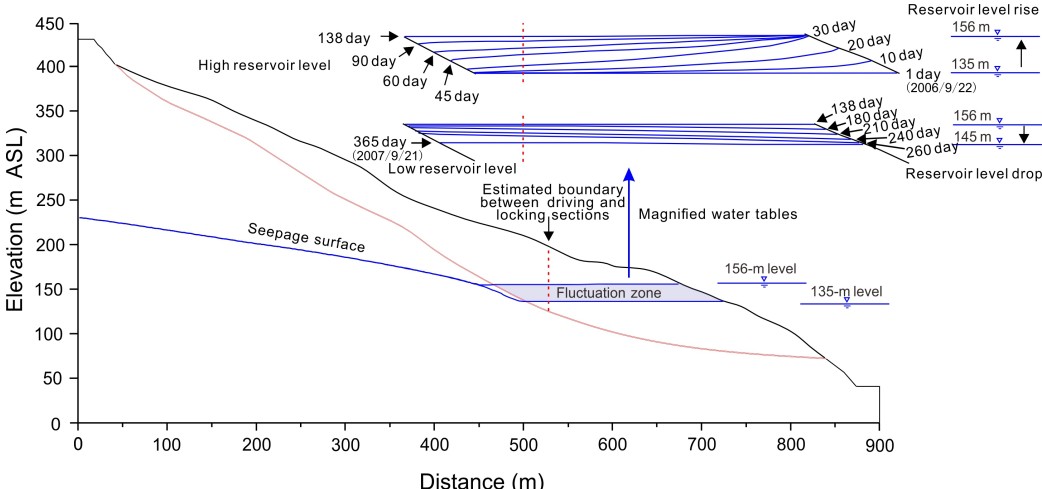

**Fig. 16** Simulated groundwater tables as the variation of reservoir water level from 22 September 2006 to 21 September 2007.

**5.3 Scenario 3: trial impoundment at 175 m *ASL***

During 2008 to 2014 the reservoir level periodically fluctuated between 145 and 175 m ASL (Stage 3), in accordance with a generalized annual water level variation curve that consists of five phases (Fig. 13(B)).

During the slow drawdown period, the groundwater storage in the driving section is reduced by an amount that approximately matches the reduction in the locking section (Fig. 17(A)), so the effect of buoyancy forces on slope stability is small. Moreover, because drawdown is slow, groundwater gradients are also low, limiting the magnitude of destabilizing seepage forces. Thus, the safety factor of the slope decreases from 1.031 to 1.018 with only a modest amount (Fig. 14(B)).

During the rapid drawdown phase, groundwater gradients are steeper and produce large, destabilizing seepage forces on the slope. The sharp decline of slope stability (Fig. 17(B)) is consistent with the observed high deformation velocity during this phase. The slope stability


becomes least (*Fos*=0.995) as the reservoir declines to its lowest level of 145 m ASL, when a
maximum difference of 14 m is computed for groundwater levels in the slide mass (Fig. 17(B)).
Although the decreased buoyancy of the locking section makes an offsetting contribution to slope
stability, its magnitude is small compared to that of destabilizing seepage forces.

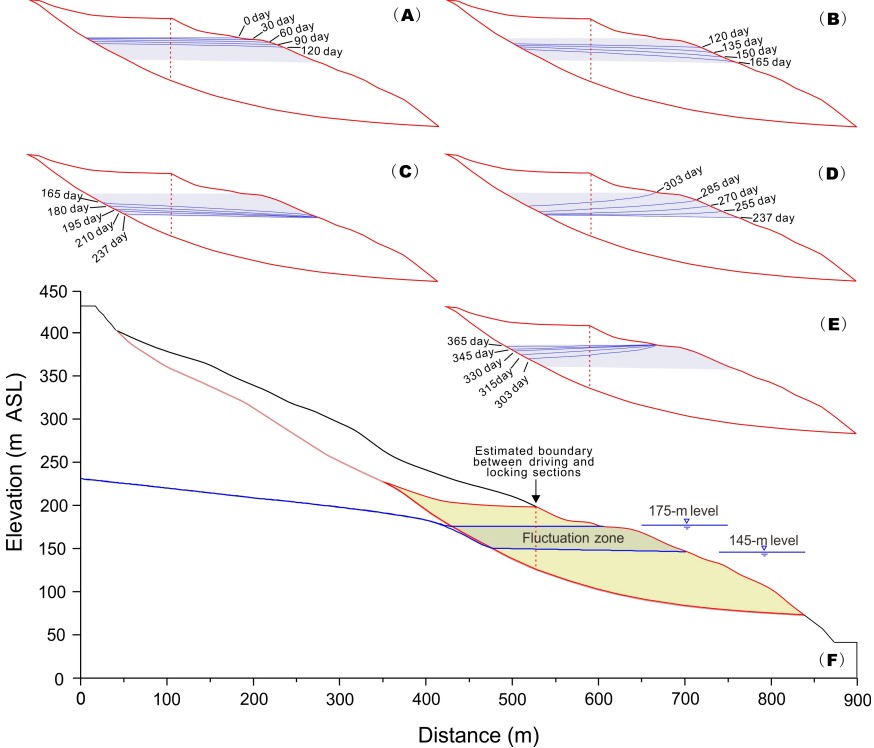


**Fig. 17** Simulated groundwater tables over the period of generalized annual variation of reservoir
water level in Stage 3. Gray shaded zone depicts the 145 to 175 m elevation interval. (A) slow
drawdown phase; (B) rapid drawdown phase; (C) low level phase; (D) water level rising phase; (E)
high water level phase

In the following three phases, representing the low water, rising and high water phases, the

characteristics of the slope vary in a manner similar to those modeled in scenario 2. The stability of





the landslide (see Fig. 14(B)) recovers gradually from 0.995 to 1.027 in the low water level phase,

due to the dissipation of destabilizing seepage forces (Fig. 17(C)). Slope stability then increases

rapidly as the reservoir level rises rapidly, when the seepage force reverses to become directed into

the slope (Fig. 17(D)). The slope obtains the highest stability with *Fos* value of 1.067 when the water

level rises to the highest level 175 m ASL. Slope stability then decreases gradually as that seepage

force declines (Fig. 17(E)). All these results agree with the observed variations in deformation

velocity of the Shuping landslide (Sec. 4.2).

In summary, during periods of reservoir drawdown and rise, the seepage force plays a dominant

role in the stability of Shuping landslide, but being negative in drawdown period and positive in the

rising period. In contrast, buoyancy effects become increasingly important during periods of steady

reservoir levels, as seepage forces steadily decrease.

## 6 Discussion

This deformation of the Shuping landslide is a function of reservoir levels but probably also

depends on the hydraulic character of its constituent material. The lower part of the slide mass that is

subject to reservoir level fluctuation is mainly composed of dense silty soil with very low hydraulic

conductivity. During periods of rapid change in reservoir level, large differences in groundwater head

can be formed in such material, generating large seepage pressures that can either destabilize or

stabilize the mass, depending on whether the reservoir is rising or falling. On the other hand, low

permeability materials impede rainfall infiltration, rendering the landslide little influenced by rainfall.

Consequently, variations of the reservoir level and their attendant seepage forces dominate the

deformation of Shuping landslide.





The evolutionary trend of the Shuping landslide under periodical water-level variations is a
significant issue. Shuping landslide has already moved horizontally by as much as ~5 m. As the mass
has descended, more material has migrated from the driving section to the locking section. The
reduction in weight of the driving section and the increased weight of the locking section has likely
improved slope stability. Support for this inference is found in the decreased deformation velocities
and decreased magnitude of the "jumps" in the observed, step-like cumulative curves in 2013 and
2014 (Fig. 10).
Based on this observation and on the results of the driving-locking model, two approaches are
recommended to control the deformation of huge reservoir landslides where the reinforcement
structures are difficult to construct. One method to improve stability is to transfer earth mass from
the driving section to the locking section of the slide mass. The other is to use drains or pumps to
lower the water levels inside the slope, in order to reduce differences in groundwater head during
periods of reservoir drawdown. The first approach has in fact been adopted to enhance the stability of
Shuping landslide. Fig. 18(A) presents the layout of the engineering treatment and Fig. 18(B) is the
subsequent photo of Shuping landslide. Zones Ⅰ and Ⅱ are the areas of load reduction, located in
the driving section of the slide mass. The earth mass of Zone Ⅰ (~1.8×10$^5$ m$^3$) and Zone Ⅱ
(~4.0×10$^5$ m$^3$) were transferred to Zones Ⅲ and Ⅳ respectively, which are located in the locking
section that is mostly below reservoir level in the photo (Fig. 18(B)). The transfer operation began in
September 2014 and was completed in June 2015. Monitoring data show that the deformation
velocity was significantly reduced to low values (about 4.1 mm/month in the main deformation area),
demonstrating the effectiveness of the engineering treatment. These approaches are more economical
and require a shorter construction period than many commonly-used remediation methods such as
the construction of stabilizing piles. Most importantly, these treatments are feasible for many other



large reservoir landslides.

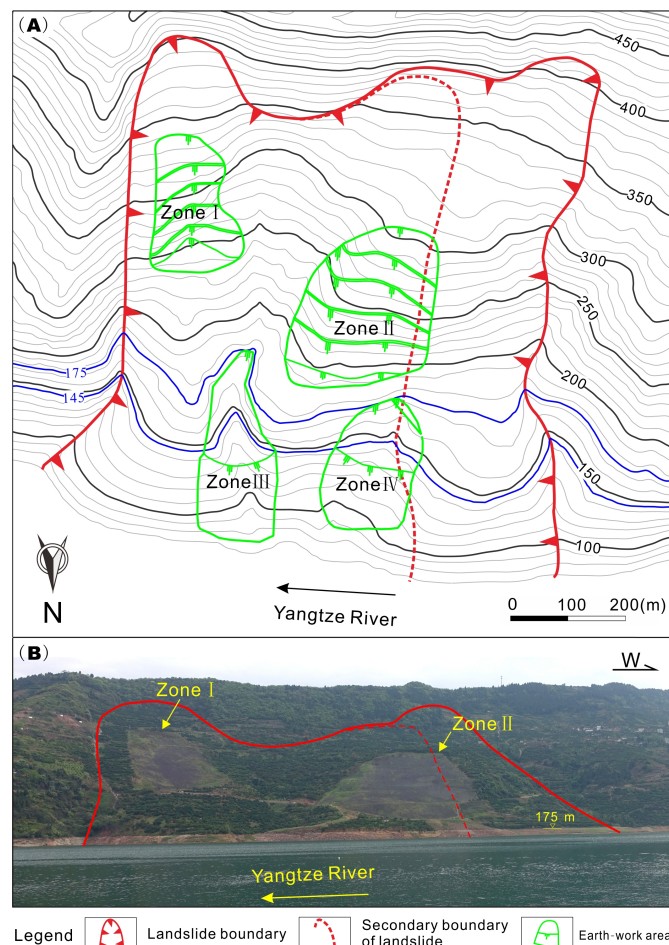


**Fig. 18** Topography of Shuping landslide before (A) and after (B) engineering treatment, which

involved the transfer of earth from Zones Ⅰ and Ⅱ to Zones Ⅲ and Ⅳ.

## 7 Conclusions

A driving-locking model is presented to elucidate the deformation mechanism of reservoir

landslides, as exemplified by Shuping landslide. The deformation velocity of Shuping landslide is
closely related to the variations in the level of the Three Gorges reservoir. Rainfall effects are limited





in comparison, perhaps due to the low hydraulic conductivity of the slide material. Rapid reservoir
drawdown produces large, destabilizing seepage forces in the slope of the slide mass, as evidenced
by large increases of its deformation velocity. In contrast, rising reservoir levels reverse the direction
of the seepage force, improving slope stability and decreasing the deformation velocity. The
buoyancy effect on the locking section decreased the slope stability when the reservoir first rose to
135 m ASL, but this effect has diminished as the reservoir has attained higher levels that buoy both
the driving and locking sections.
Monitoring data, the driving-locking model, and a successful engineering treatment suggest two
means to increase the stability of landslides in the TGR area. Recommended approaches are: 1)
transferring earth mass from the driving section to the locking section; and 2) lowering the ground
water levels inside the slope by drains or by pumping during periods of reservoir drawdown. The
first approach was successfully applied to the Shuping landslide and could be used to treat many
other huge landslides in the Three Gorges Reservoir area.
**Data availability**
The study relied on the observation data from Department of Land and Resources of Hubei
Province, China.
**Competing interests**
The authors declare that they have no conflict of interest.





**Acknowledgements**
This work was supported by the National Key R&D Program of China (No. 2017YFC1501305);
the Fundamental Research Funds for the Central Universities, China University of Geosciences
(Wuhan) (No. CUGCJ1701); and the National Natural Science Foundation of China (Nos. 41630643,

41827808, 41502290).




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
