# Peer review of "A model for interpreting the deformation mechanism of reservoir landslides in the"

_Natural Hazards and Earth System Sciences, 2019_

## Referee Comment (RC1) · Anonymous Referee #1 · 8 Jun 2020

General comments:

As a whole, the manuscript is valuable and presents robust data for publication. However, some parts of the manuscipt are completely useless and uncorrect from a theorical point of view, while some other parts require modifications. Therefore, this reviewer suggests a strong re-structuring of the manuscript as well as an improvement of the parts that need corrections. English is generally fine and no significant typing errors have been detected. Here follows some of the main revisions required:

- In the introduction section, the authors should better describe, from a theorical point of view, the problem of rapid drawdown and rainfall infiltration in the landslide equilibrium,

and in particular the role of permeability of the landslide soils and the rate of drawdown. Is this problem related to the type of the soils involved or not?

- In the driving-locking model (Section 2), the authors do not completely account for the general equilibrium of the landslide mass, since they reduce all the equilibrium condition to the single unit vertical slice without considering the inter-slice forces, which do have a role in the equilibrium of the single slice. This is uncorrect, since it affects the location of the locking section. All this section, and the equations here proposed, seems to be a neglection of the slice methods historically proposed in the limit equilibrium approach and in general of the equilibrium theory (the problem being undetermined from a statical point of view and the need of integrative equations to balance unknowns-equations...). Moreover, in the limit equilibrium analysis proposed by the authors in the following sections, they use the Morgenstern-Price method, which is a well-known rigorous method and of course takes into account the inter-slice forces. Therefore, the first part of the manuscript is not in agreement with the approach followed in the second part. This reviewer suggests to completely remove Section 2 from the manuscript and eventually to extend the second part (seepage and LE analysis) by including new field or analytical data and relative discussion.

- The distinction between driving section and locking section (I would suggest "resisting section" rather than "locking", if necessary) is not rigorous and can have only a qualitative meaning. Even in the driving section, there is some mobilised strength component along the corresponding portion of the sliding surface, as well as even in the locking section the driving forces, in some circumstances, can prevail over the resisting ones.

Specific comments:

- In the figures proposed the term "deformations" is used to indicate displacements, which have mm as measurement unit. Please, use the term "displacements". - The comment presented at lines 456-461 is questionable, since a displacement of 5 m is not so large to justify a a change in the landslide body geometry, especially for a landslide size as that here examined. Apart from the change in the curve trends, a limit equilibrium analysis with the post-movement landslide geometry should be performed to verify the actual change in the factor of safety. - The cohesion value adopted for the sliding surface should be justified more in detail. The landslide is moving and has experienced quite a large displacement; therefore, probably the cohesion value proposed is not operative anymore and, in general, post-failure strength conditions would apply in this situation. A comment from the authors on this choice is necessary. - A more detailed description of the engineering treatment performed in the slope is necessary. It is mentioned, but not described. - Since a transient seepage analysis is carried out, the authors should describe also some more data on the hydraulic properties of the soils used in the seepage calculations, as required by the software code used (retention curves, permeability coefficient variation with suctions). - Line 338: what does it exactly mean "rainfall threshold" as expressed in terms of rainfall intensity? Being clay materials, rainfall data in terms of long-term cumulative rainfalls should be more important than rainfall intensity. - Dam impoundment has also an external loading (i.e.stabilizing) function on the landslide equilibrium. The external impoundment load affects the overall equilibrium of the landslide body. This is never mentioned by the authors. - Since the authors explain the change in the equilibrium conditions of the landslide in terms of seepage forces (inward or outward, with respect to the slope), they should plot the output of the seepage analysis in terms of flow vectors (during a drawdown stage and an impoundment stage, for example) in order to corroborate their comments. - How is chosen the location of the section dividing the driving and locking portions based on the results of the analyses proposed?

---

## Referee Comment (RC2) · Anonymous Referee #2 · 21 Jul 2020

The subject manuscript, "A model for interpreting the deformation mechanism of reservoir landslides in the Three Gorges Reservoir area, China" is an important case study of a large, deep landslide that has been affected by reservoir impoundment and fluctuations. The manuscript is logically organized, well written and presents a long record of data relating landslide movement, reservoir levels, and precipitation.

My primary criticism of the paper is that the authors seem to be unaware of previous studies that have presented similar, closely related models to that presented in sections 2.2 and 2.3. Although most previous work cited in the following lines does not specifically address reservoir effects on landslides, the relationships between landslide geometry, deformation, dynamics, and stability identified in previous studies is relevant to the case presented in the subject manuscript. The model has concepts in common with the wedge method for analyzing landslides consisting of an active driving wedge and resisting block (Terzaghi & Peck, 1967; Sultan and Seed, 1967). Hutchinson (1984) presented an "influence-line" approach for assessing effectiveness of cuts and fills in stabilizing slopes, which is also similar to the models in sections 2.2 and 2.3. Iverson (1986) described relationships between stress distribution and landslide geometry. Baum and Fleming (1991) described the relationship between displacement patterns and the results of stability analysis, and derived expressions for the boundary between driving and resisting elements of landslides. Interestingly, they concluded that the boundary is near the thickest part of the landslide, consistent with the findings of this manuscript. Drawing on insights gained from these earlier studies, McKean and Roering (2004), Guerriero et al. (2014), Prokesova et al. (2014), and Handwerger et al. (2015) as well as others, have further explored the influence of slip-surface and landslide geometry on slide deformation, force distribution and landslide dynamics.

In addition to strengthening the background section/literature review to show the relationship of the authors' model to previous work,

References cited:

Baum, R.L., Fleming, R.W., 1991. Use of longitudinal strain in identifying driving and resisting elements of landslides. Geol. Soc. Am. Bull. 103, 1121–1132.

Guerriero, L., Coe, J.A., Revellino, P., Grelle, G., Pinto, F., and Guadagno, F.M., 2014, Influence of slip-surface geometry on earth-flow deformation, Montaguto earth flow, southern Italy: Geomorphology, v. 219, p. 285-305. http://dx.doi.org/10.1016/j.geomorph.2014.04.039

Handwerger, A.L., Roering, J., Schmidt, D.A., and Rempel, A.W., 2015, Kinematics of earthflows in the Northern California Coast Ranges using satellite interferometry: Geomorphology v. 246, p.321–333.

Hutchinson, J.N., 1984, An influence line approach to the stabilization of slopes by cuts and fills: Canadian Geotechnical Journal, v. 21, p. 363-370.

Iverson, R.M., 1986. Unsteady, nonuniform landslide motion: 2. Linearized theory and the kinematics of transient response. J. Geol. 349–364.

McKean, J. and Roering, J. 2004, Objective landslide detection and surface morphology mapping using high-resolution airborne laser altimetry: Geomorphology 57 (2004) 331–351

Prokešová, R., Kardoš, M., Tábork, P., Medvedová, A., Stacke, V., Chudy, F., 2014. Kinematic behaviour of a large earthflow defined by surface displacement monitoring, dem differencing, and ert imaging. Geomorphology 224, 86–101.

Sultan, H.A., and Seed, H.B., 1967, Stability of sloping core earth dams: American Society of Civil Engineers Proceedings, Journal of the Soil Mechanics and Foundations Division, V. 93, no. SM4, p. 45-68.

Terzaghi, K. and Peck, R.B., 1967, Soil mechanics in engineering practice (2nd ed.): New York, Wiley, 729 p.
* * *

---

## Author Comment (AC1) · 29 Jul 2020

Dear Editor and Reviewer, Thank you for editor's efforts on dealing our manuscript and reviewer's constructive comments on this manuscript. We have studied these comments carefully and made point-by-point corrections, which have enabled us to improve the manuscript. Now we present point-by-point response to reviewer's comments, followed by the revised manuscript. The revised portions are marked in RED in new manuscript (MS). Below we list every comment received (in italics), followed by our response in regular font.

General comment 1. As a whole, the manuscript is valuable and presents robust data

for publication. However, some parts of the manuscript are completely useless and un-correct from a theoretical point of view, while some other parts require modifications. Therefore, this reviewer suggests a strong re-structuring of the manuscript as well as an improvement of the parts that need corrections. English is generally fine and no significant typing errors have been detected. Response: Thanks for reviewer's comments and suggestions. 2. In the introduction section, the authors should better describe, from a theorical point of view, the problem of rapid drawdown and rainfall infiltration in the landslide equilibrium, and in particular the role of permeability of the landslide soils and the rate of drawdown. Is this problem related to the type of the soils involved or not? Response: Yes, the problem is related to the type of the soils; landslides with lower permeability are more susceptible to be affected by the drawdown. We now add content as reviewer suggested to describe the effect of rapid drawdown and landslide permeability on landslide stability (new Lines: 61-65). The added contents are as be-low: These phenomena are more obvious in the landslides with lower permeability and in the situations of rapid drawdown and heavy rainfall. In the low permeability land-slide, the groundwater is not easy to be discharged from the slope in the process of rapid drawdown and rainfall infiltration, which results in the formation of pressure dif-ference between inside and outside of the landslide and reduces the stability of the landslide. 3. In the driving-locking model (Section 2), the authors do not completely account for the general equilibrium of the landslide mass, since they reduce all the equilibrium condition to the single unit vertical slice without considering the inter-slice forces, which do have a role in the equilibrium of the single slice. This is uncorrect, since it affects the location of the locking section. All this section, and the equations here proposed, seems to be a neglection of the slice methods historically proposed in the limit equilibrium approach and in general of the equilibrium theory (the problem being undetermined from a statical point of view and the need of integrative equations to balance unknowns- equations. . .). Moreover, in the limit equilibrium analysis pro-posed by the authors in the following sections, they use the Morgenstern-Price method, which is a well-known rigorous method and of course takes into account the inter-slice forces. Therefore, the first part of the manuscript is not in agreement with the approach followed in the second part. This reviewer suggests to completely remove Section 2 from the manuscript and eventually to extend the second part (seepage and LE analysis) by including new field or analytical data and relative discussion. Response: Yes, the limit equilibrium method has developed from simplified limit equilibrium methods to rigorous limit equilibrium (LE) methods, and we agree with reviewer that a rigorous LE method would give more precise result about the location of the locking section. But we still choose the simplified limit equilibrium method for analysis here for following reasons. The locking section is defined as the lower-front part of the slide mass, where each unit vertical slice (Fig. 3) can be self-stabilized under its self-weight. In the unit vertical slice of locking section, the difference between the forces on the two vertical sides is very tiny because the width of the unit vertical slice is very small, and the slide surface underlying the lower-front part of the slide mass is relatively gentle; so the interslice forces were ignored for convenience of analysis. Moreover, the second reviewer says "Interestingly, they concluded that the boundary is near the thickest part of the landslide, consistent with the findings of this manuscript", which demonstrates that our used LE method here is acceptable. So we insist to preserve the Section 2. In the Section 5, the rigorous limit equilibrium method (M-P method) is employed to analyze the Shuping landslide, which is not consistent with that used in the Section 2. Because we want to use rigorous LE to check the results from the simplified LE method used in the Section 2. To address this comment, we added an explanation to clarify why we choose the simplified LE method in the section 2 (on Lines: 124-127). 4. The distinction between driving section and locking section (I would suggest "resisting section" rather than "locking", if necessary) is not rigorous and can have only a qualitative meaning. Even in the driving section, there is some mobilised strength component along the corresponding portion of the sliding surface, as well as even in the locking section the driving forces, in some circumstances, can prevail over the resisting ones. Response: We agree with reviewer's opinion. We now change the term "locking section" into "resisting section" in the whole manuscript as suggested.

Specific comments: 1. In the figures proposed the term "deformations" is used to indicate displacements, which have mm as measurement unit. Please, use the term "displacements". Response: Thanks for catching this error. "deformation" was changed to "displacement" in the new MS (Figure 12). 2. The comment presented at lines 456-461 is questionable, since a displacement of 5 m is not so large to justify a change in the landslide body geometry, especially for a landslide size as that here examined. Apart from the change in the curve trends, a limit equilibrium analysis with the post-movement landslide geometry should be performed to verify the actual change in the factor of safety. Response: The accurate calculation of the safety factor of the landslide with the change of the landslide body geometry is unavailable here, because the accurate post-movement landslide geometry is difficult to be obtained. To address this comment, we removed this questionable content.

3. The cohesion value adopted for the sliding surface should be justified more in detail. The landslide is moving and has experienced quite a large displacement; therefore, probably the cohesion value proposed is not operative anymore and, in general, post-failure strength conditions would apply in this situation. A comment from the authors on this choice is necessary. Response: We agree with reviewer's opinion. Shuping landslide is a reactivated landslide and had experienced large deformation before the reservoir impoundment; therefore the post-failure strength was applied in the calculation in this study. 4. A more detailed description of the engineering treatment performed in the slope is necessary. It is mentioned, but not described. Response: Thanks for reminding. We presented the detailed description of the engineering treatment in the Section 6 (on Lines: 480-486). 5. Since a transient seepage analysis is carried out, the authors should describe also some more data on the hydraulic properties of the soils used in the seepage calculations, as required by the software code used (retention curves, permeability coefficient variation with suctions). Response: Thanks for reminding. We added the necessary hydraulic properties in Tab 1. 6. Line 338: what does it exactly mean "rainfall threshold" as expressed in terms of rainfall intensity? Being clay materials, rainfall data in terms of long-term cumulative rainfalls should be more important than rainfall intensity. Response: Yes, the "rainfall threshold" is expressed in the terms of the monthly rainfall here, which represents monthly cumulative rainfall. 7. Dam impoundment has also an external loading (i.e.stabilizing) function on the landslide equilibrium. The external impoundment load affects the overall equilibrium of the landslide body. This is never mentioned by the authors. Response: The external impoundment load affect has been considered within the SLOPE/W module of GEOSTUDIO software. To address this comment, we mentioned this factor in the new MS (on Lines: 366-367) 8. Since the authors explain the change in the equilibrium conditions of the landslide in terms of seepage forces (inward or outward, with respect to the slope), they should plot the output of the seepage analysis in terms of flow vectors (during a drawdown stage and an impoundment stage, for example) in order to corroborate their comments. Response: It needs a lot of space to present the flow vectors in the whole process of drawdown stage and impoundment stage, because in the every state, it needs a separated figure. While, the phreatic lines, which is closely relevant to the seepage force in the LE analysis, can be overlap displayed and reflect the whole process in one figure. Therefore, the phreatic lines are still used here. 9. How is chosen the location of the section dividing the driving and locking portions based on the results of the analyses proposed? Response: We analyzed this issue in Section 2.2, and the conclusion is that the boundary between the locking and driving sections can be approximated as the position where the slope angle $\theta_1$ equals the internal friction angle $\varphi$ ( on Lines 1489-157).

Thanks again for editor's and reviewer's effort on our manuscript! Best regards,

Zongxing Zou, Huiming Tang, Robert E. Criss, Xinli Hu, Chengren Xiong, Qiong Wu, Yi Yuan

Please also note the supplement to this comment:
https://nhess.copernicus.org/preprints/nhess-2019-432/nhess-2019-432-AC1-supplement.pdf

[Figure]

**Supplement:**

**Reply to Reviewer 1's comments on *"A model for interpreting the deformation mechanism of reservoir landslides in the Three Gorges Reservoir area, China"* (nhess-2019-432)**

Dear Editor and Reviewer,

Thank you for editor's efforts on dealing our manuscript and reviewer's constructive comments on this manuscript. We have studied these comments carefully and made point-by-point corrections, which have enabled us to improve the manuscript. Now we present point-by-point response to reviewer's comments, followed by the revised manuscript. The revised portions are marked in RED in new manuscript (MS).

Below we list every comment received (in *italics*), followed by our response in regular font.

**General comment**

1. *As a whole, the manuscript is valuable and presents robust data for publication. However, some parts of the manuscript are completely useless and uncorrect from a theoretical point of view, while some other parts require modifications. Therefore, this reviewer suggests a strong re-structuring of the manuscript as well as an improvement of the parts that need corrections. English is generally fine and no significant typing errors have been detected.*

**Response:** Thanks for reviewer's comments and suggestions.

2. *In the introduction section, the authors should better describe, from a theorical point of view, the problem of rapid drawdown and rainfall infiltration in the landslide equilibrium, and in particular the role of permeability of the landslide soils and the rate of drawdown. Is this problem related to the type of the soils involved or not?*

**Response:** Yes, the problem is related to the type of the soils; landslides with lower permeability are more susceptible to be affected by the drawdown. We now add content as reviewer suggested to describe the effect of rapid drawdown and landslide permeability on landslide stability (new Lines: 61-65).

The added contents are as below: *These phenomena are more obvious in the landslides with lower permeability and in the situations of rapid drawdown and heavy rainfall. In the low permeability landslide, the groundwater is not easy to be discharged from the slope in the process of rapid drawdown and rainfall infiltration, which results in the formation of pressure difference between inside and outside of the landslide and reduces the stability of the landslide.*

3. *In the driving-locking model (Section 2), the authors do not completely account for the general equilibrium of the landslide mass, since they reduce all the equilibrium condition to the single unit vertical slice without considering the inter-slice forces, which do have a role in the equilibrium of the single slice. This is uncorrect, since it affects the location of the locking section. All this section, and the equations here proposed, seems to be a neglection of the slice methods historically proposed in the limit equilibrium approach and in general of the equilibrium theory (the problem being undetermined from a statical point of view and the need of integrative equations to balance unknowns- equations. . .). Moreover, in the limit equilibrium analysis proposed by the authors in the following sections, they use the Morgenstern-Price method, which is a well-known rigorous method and of course takes into account the inter-slice forces. Therefore, the first part of the manuscript is not in agreement with the approach followed in the second part. This reviewer suggests to completely remove Section 2 from the manuscript and eventually to extend the second part (seepage and LE analysis) by including new field or analytical data and relative discussion.*

**Response:** Yes, the limit equilibrium method has developed from simplified limit equilibrium methods to rigorous limit equilibrium (LE) methods, and we agree with reviewer that a rigorous LE method would give more precise result about the location of the locking section. But we still choose the simplified limit equilibrium method for analysis here for following reasons. The locking section is defined as the lower-front part of the slide mass, where each unit vertical slice (Fig. 3) can be self-stabilized under its self-weight. In the unit vertical slice of locking section, the difference between the forces on the two vertical sides is very tiny because the width of the unit vertical slice is very small, and the slide surface underlying the lower-front part of the slide mass is relatively gentle; so the interslice forces were ignored for convenience of analysis. Moreover, the second reviewer says "Interestingly, they concluded that the boundary is near the thickest part of the landslide, consistent with the findings of this manuscript", which demonstrates that our used LE method here is acceptable. So we insist to preserve the Section 2.

In the Section 5, the rigorous limit equilibrium method (M-P method) is employed to analyze the Shuping landslide, which is not consistent with that used in the Section 2. Because we want to use rigorous LE to check the results from the simplified LE method used in the Section 2.

To address this comment, we added an explanation to clarify why we choose the simplified LE method in the section 2 (on Lines: 124-127).

4. ***The distinction between driving section and locking section (I would suggest "resisting section" rather than "locking", if necessary) is not rigorous and can have only a qualitative meaning. Even in the driving section, there is some mobilised strength component along the corresponding portion of the sliding surface, as well as even in the locking section the driving forces, in some***

*circumstances, can prevail over the resisting ones.*

**Response:** We agree with reviewer's opinion. We now change the term "locking section" into "resisting section" in the whole manuscript as suggested.

**Specific comments:**

1. *In the figures proposed the term "deformations" is used to indicate displacements, which have mm as measurement unit. Please, use the term "displacements".*

**Response:** Thanks for catching this error. "deformation" was changed to "displacement" in the new MS (Figure 12).

2. *The comment presented at lines 456-461 is questionable, since a displacement of 5 m is not so large to justify a change in the landslide body geometry, especially for a landslide size as that here examined. Apart from the change in the curve trends, a limit equilibrium analysis with the post-movement landslide geometry should be performed to verify the actual change in the factor of safety.*

**Response:** The accurate calculation of the safety factor of the landslide with the change of the landslide body geometry is unavailable here, because the accurate post-movement landslide geometry is difficult to be obtained. To address this comment, we removed this questionable content.

3. *The cohesion value adopted for the sliding surface should be justified more in detail. The landslide is moving and has experienced quite a large displacement; therefore, probably the cohesion value proposed is not operative anymore and, in general, post-failure strength conditions would apply in this situation. A comment from the authors on this choice is necessary.*

**Response:** We agree with reviewer's opinion. Shuping landslide is a reactivated landslide and had experienced large deformation before the reservoir impoundment; therefore the post-failure strength was applied in the calculation in this study.

4. *A more detailed description of the engineering treatment performed in the slope is necessary. It is mentioned, but not described.*

**Response:** Thanks for reminding. We presented the detailed description of the engineering treatment in the Section 6 (on Lines: 480-486).

5. *Since a transient seepage analysis is carried out, the authors should describe also some more data on the hydraulic properties of the soils used in the seepage calculations, as required by the software code used (retention curves, permeability coefficient variation with suctions).*

**Response:** Thanks for reminding. We added the necessary hydraulic properties in Tab 1.

***6.  Line 338:  what does it exactly mean "rainfall threshold" as expressed in terms of rainfall intensity? Being clay materials, rainfall data in terms of long-term cumulative rainfalls should be more important than rainfall intensity.***

**Response:** Yes, the "rainfall threshold" is expressed in the terms of the monthly rainfall here, which represents monthly cumulative rainfall.

***7.  Dam impoundment has also an external loading (i.e.stabilizing) function on the landslide equilibrium. The external impoundment load affects the overall equilibrium of the landslide body. This is never mentioned by the authors.***

**Response:** The external impoundment load affect has been considered within the SLOPE/W module of GEOSTUDIO software. To address this comment, we mentioned this factor in the new MS (on Lines: 366-367)

***8.  Since the authors explain the change in the equilibrium conditions of the landslide in terms of seepage forces (inward or outward, with respect to the slope), they should plot the output of the seepage analysis in terms of flow vectors (during a drawdown stage and an impoundment stage, for example) in order to corroborate their comments.***

**Response:** It needs a lot of space to present the flow vectors in the whole process of drawdown stage and impoundment stage, because in the every state, it needs a separated figure. While, the phreatic lines, which is closely relevant to the seepage force in the LE analysis, can be overlap displayed and reflect the whole process in one figure. Therefore, the phreatic lines are still used here.

***9.  How is chosen the location of the section dividing the driving and locking portions based on the results of the analyses proposed?***

**Response:** We analyzed this issue in Section 2.2, and the conclusion is that the boundary between the locking and driving sections can be approximated as the position where the slope angle $\theta_1$ equals the internal friction angle $\varphi$ ( on Lines 1489-157).

Thanks again for editor's and reviewer's effort on our manuscript!

Best regards,

[revised manuscript text omitted]

---

## Author Comment (AC2) · 29 Jul 2020

Dear Editor and Reviewer, Thank you for editor's efforts on dealing our manuscript and reviewer's very kind comment on your manuscript. We have studied reviewer's comments carefully and made corrections as suggested. The revised portions are marked in RED in new manuscript (MS). Below we list every comment received (in italics), followed by our response in regular font.

General comment 1. The subject manuscript, "A model for interpreting the deformation mechanism of reservoir landslides in the Three Gorges Reservoir area, China" is an important case study of a large, deep landslide that has been affected by reservoir impoundment and fluctuations. The manuscript is logically organized, well written and presents a long record of data relating landslide movement, reservoir levels, and precipitation. Response: Thanks for reviewer's kind comments.

Specific comments: 1. My primary criticism of the paper is that the authors seem to be unaware of previous studies that have presented similar, closely related models to that presented in sections 2.2 and 2.3. Although most previous work cited in the following lines does not specifically address reservoir effects on landslides, the relationships between landslide geometry, deformation, dynamics, and stability identified in previous studies is relevant to the case presented in the subject manuscript. The model has concepts in common with the wedge method for analyzing landslides consisting of an active driving wedge and resisting block (Terzaghi & Peck, 1967; Sultan and Seed, 1967). Hutchinson (1984) presented an "influence-line" approach for assessing effectiveness of cuts and fills in stabilizing slopes, which is also similar to the models in sections 2.2 and 2.3. Iverson (1986) described relationships between stress distribution and landslide geometry. Baum and Fleming (1991) described the relationship between displacement patterns and the results of stability analysis, and derived expressions for the boundary between driving and resisting elements of landslides. Interestingly, they concluded that the boundary is near the thickest part of the landslide, consistent with the findings of this manuscript. Drawing on insights gained from these earlier studies, McKean and Roering (2004), Guerriero et al. (2014), Prokesova et al. (2014), and Handwerger et al. (2015) as well as others, have further explored the influence of slip-surface and landslide geometry on slide deformation, force distribution and landslide dynamics. In addition to strengthening the background section/literature review to show the rela- tionship of the authors' model to previous work. References cited: ïČij Baum, R.L., Fleming, R.W., 1991. Use of longitudinal strain in identifying driving and resisting elements of landslides. Geol. Soc. Am. Bull. 103, 1121–1132. ïČij Guerriero, L., Coe, J.A., Revellino, P., Grelle, G., Pinto, F., and Guadagno, F.M., 2014, Influence of slip-surface geometry on earth-flow deformation, Mon- taguto earth flow, southern Italy: Geomorphology, v. 219, p. 285-305.

http://dx.doi.org/10.1016/j.geomorph.2014.04.039 ïČij Handwerger, A.L., Roering, J., Schmidt, D.A., and Rempel, A.W., 2015, Kinematics of earthflows in the Northern California Coast Ranges using satellite interferometry: Geomorphology v. 246, p.321–333. ïČij Hutchinson, J.N., 1984, An influence line approach to the stabilization of slopes by cuts and fills: Canadian Geotechnical Journal, v. 21, p. 363-370. ïČij Iverson, R.M., 1986. Unsteady, nonuniform landslide motion: 2. Linearized theory and the kinematics of transient response. J. Geol. 349–364. ïČij McKean, J. and Roering, J. 2004, Objective landslide detection and surface morphol- ogy mapping using high-resolution airborne laser altimetry: Geomorphology 57 (2004) 331–351 ïČij Prokešová, R., Kardoš, M., Tábork, P., Medvedová, A., Stacke, V., Chudy, F., 2014. Kinematic behaviour of a large earthflow defined by surface displacement monitoring, dem differencing, and ert imaging. Geomorphology 224, 86–101. ïČij Sultan, H.A., and Seed, H.B., 1967, Stability of sloping core earth dams: American Society of Civil Engineers Proceedings, Journal of the Soil Mechanics and Foundations Division, V. 93, no. SM4, p. 45-68. ïČij Terzaghi, K. and Peck, R.B., 1967, Soil mechanics in engineering practice (2nd ed.): New York, Wiley, 729 p.

Response: Many thanks for reviewer providing these valuable references. We now add a background section to review these references and address the relationship between our work and the previous work (on Lines: 71-79).

Thanks again for editor's and reviewer's effort on our manuscript! Best regards,

Zongxing Zou, Huiming Tang, Robert E. Criss, Xinli Hu, Chengren Xiong, Qiong Wu, Yi Yuan

Please also note the supplement to this comment:
https://nhess.copernicus.org/preprints/nhess-2019-432/nhess-2019-432-AC2-supplement.pdf

―――――――――――――――――――――

2019-432, 2020.

**Supplement:**

**Reply to Reviewer 2's comments on *"A model for interpreting the deformation mechanism of reservoir landslides in the Three Gorges Reservoir area, China"* (nhess-2019-432)**

Dear Editor and Reviewer,

Thank you for editor's efforts on dealing our manuscript and reviewer's very kind comment on your manuscript. We have studied reviewer's comments carefully and made corrections as suggested. The revised portions are marked in RED in new manuscript (MS).

Below we list every comment received (in *italics*), followed by our response in regular font.

**General comment**

1. *The subject manuscript, "A model for interpreting the deformation mechanism of reservoir landslides in the Three Gorges Reservoir area, China" is an important case study of a large, deep landslide that has been affected by reservoir impoundment and fluctuations. The manuscript is logically organized, well written and presents a long record of data relating landslide movement, reservoir levels, and precipitation.*

**Response:** Thanks for reviewer's kind comments.

**Specific comments:**

1. *My primary criticism of the paper is that the authors seem to be unaware of previous studies that have presented similar, closely related models to that presented in sections 2.2 and 2.3. Although most previous work cited in the following lines does not specifically address reservoir effects on landslides, the relationships between landslide geometry, deformation, dynamics, and stability identified in previous studies is relevant to the case presented in the subject manuscript. The model has concepts in common with the wedge method for analyzing landslides consisting of an active driving wedge and resisting block (Terzaghi & Peck, 1967; Sultan and Seed, 1967). Hutchinson (1984) presented an "influence-line" approach for assessing effectiveness of cuts and fills in stabilizing slopes, which is also similar to the models in sections 2.2 and 2.3. Iverson (1986) described relationships between stress distribution and landslide geometry. Baum and Fleming (1991) described the relationship between displacement patterns and the results of stability analysis, and derived expressions for the boundary between driving and resisting elements of landslides.*

*Interestingly, they concluded that the boundary is near the thickest part of the landslide, consistent with the findings of this manuscript. Drawing on insights gained from these earlier studies, McKean and Roering (2004), Guerriero et al. (2014), Prokesova et al. (2014), and Handwerger et al. (2015) as well as others, have further explored the influence of slip-surface and landslide geometry on slide deformation, force distribution and landslide dynamics.*

*In addition to strengthening the background section/literature review to show the rela- tionship of the authors' model to previous work.*

*References cited:*

✓ *Baum, R.L., Fleming, R.W., 1991. Use of longitudinal strain in identifying driving and resisting elements of landslides. Geol. Soc. Am. Bull. 103, 1121–1132.*

✓ *Guerriero, L., Coe, J.A., Revellino, P., Grelle, G., Pinto, F., and Guadagno, F.M., 2014, Influence of slip-surface geometry on earth-flow deformation, Mon- taguto earth flow, southern Italy: Geomorphology, v. 219, p. 285-305. http://dx.doi.org/10.1016/j.geomorph.2014.04.039*

✓ *Handwerger, A.L., Roering, J., Schmidt, D.A., and Rempel, A.W., 2015, Kinematics of earthflows in the Northern California Coast Ranges using satellite interferometry: Geomorphology v. 246, p.321–333.*

✓ *Hutchinson, J.N., 1984, An influence line approach to the stabilization of slopes by cuts and fills: Canadian Geotechnical Journal, v. 21, p. 363-370.*

✓ *Iverson, R.M., 1986. Unsteady, nonuniform landslide motion: 2. Linearized theory and the kinematics of transient response. J. Geol. 349–364.*

✓ *McKean, J. and Roering, J. 2004, Objective landslide detection and surface morphol- ogy mapping using high-resolution airborne laser altimetry: Geomorphology 57 (2004) 331–351*

✓ *Prokešová, R., Kardoš, M., Tábork, P., Medvedová, A., Stacke, V., Chudy, F., 2014. Kinematic behaviour of a large earthflow defined by surface displacement monitoring, dem differencing, and ert imaging. Geomorphology 224, 86–101.*

✓ *Sultan, H.A., and Seed, H.B., 1967, Stability of sloping core earth dams: American Society of Civil Engineers Proceedings, Journal of the Soil Mechanics and Foundations Division, V. 93, no. SM4, p. 45-68.*

✓ *Terzaghi, K. and Peck, R.B., 1967, Soil mechanics in engineering practice (2nd ed.): New York, Wiley, 729 p.*

**Response:** Many thanks for reviewer providing these valuable references. We now add a background section to review these references and address the relationship between our work and the previous work (on Lines: 71-79).

Thanks again for editor's and reviewer's effort on our manuscript!

Best regards,

[revised manuscript text omitted]

---

## Author Response (AR2)

**Response to Reviewers' comments on *"A model for interpreting the deformation mechanism of reservoir landslides in the Three Gorges Reservoir area, China"* (nhess-2019-432)**

Dear Editor and Reviewers,

Thank you for editor's efforts on dealing our manuscript and the comments from the reviewers. In this revised version, we made point-by-point corrections after carefully studying the third and fourth reviewer's comments. The revised portions are marked in RED in new manuscript (MS).

Below we list every comment received (in *italics*), followed by our response in regular font.

**Response to Reviewer 3 comments**

**General comment**

*The paper represents a methodological work based on monitoring activity of rainfall and displacements and on stability calculations of a case study at Shuping area, China. The paper is well written and clearly explain how the seepage force works during the water rising and drop level the considered reservoir. Nonetheless, some minor revisions could improve some points.*

**Response:** Thanks for reviewer's kind comments and suggestions.

**Specific comments:**

1. *Figure 1: the landslide examples should be better characterized by adding the friction angle and cohesion values, if possible.*

**Response:** Thanks for suggestion. The friction angle and cohesion values were added for the presented landslide cases in Figure 1. The friction value of the slip surface of the Vajont landslide was back-analyzed in many literatures according to its failure process, but the cohesion value was not presented. Therefore, only friction value was presented for Vajont landslide in Figure 1 (Line 84).

2. *Pag. 9, line 141-142: the internal friction angle range of values must be related to specific formations set in the studied area and the Vajont landslide must be considered separately.*

**Response:** The statistic results show that the internal friction of the Vajont landslide (Figure 1, Line 84) is in the range of the internal friction angle of the landslides in the

studied area; therefore, Vajont landslide was not considered separately.

*3. Fig. 10: the caption refers to 2013 and 2016 years, but in the abscissa it is written the beginning of 2017.*

**Response:** OK, we removed the date of the 2017 from the abscissa in the new version. Please see Figure 9 (Line 265) in new manuscript.

*4. pag. 25, line 373-376: why were these boundary conditions assigned?*

**Response:** Thanks for suggestion; the expression of the boundary conditions were simplified (Lines: 372-373).

**Response to Reviewer 4 comments**

**General comment**

*I found this manuscript is important and understandable. However, it has concerns in particular, the discussion part is too short for a scientific paper. The authors can compare other studies citing literatures. If it is a lack of profound discussion, the manuscript would be inappropriate as a scientific paper of natural hazards. For example, how does the authors consider the effect of frictional property on sliding plane. The sliding plane they showed in the profile of Shuping landslide moved on jaggy rock surface, and that asperity surely affect the mass balance of landslide. Any way I hope the authors will add fulfilling discussions with comparing other study cases, so I would recommend it to be the major revision.*

**Response:** Thanks for reviewer's kind comments and suggestions. We extended the discussion and compared the results with the references, see Lines 505-519.

**Specific comments:**

*1. Line 39-40, The authors wrote about the government expense of mitigation work, but I think it is inappropriate for the scientific paper.*

**Response:** Thanks for pointing this issue; we removed this sentence in the revised version (Lines 39-40).

*2. Fig.1 What indicates "degrees" on upper left upper of each figure? Explain in the caption. And, what means water levels?*

**Response:** "degrees" on upper left upper of each figure indicates the *section orientation*. We now add the legend in Figure 1. "water level" in Figure 1 indicates the *important reservoir level* for each landslide; we also add the legend in Figure 1.

*3. Line 141, How does the authors decide the empirical values? Please explain in*

*detail.*

**Response:** We decide the empirical values according to the range of the shear strength parameters of the slip zone soil presented in *Engineering Geology Manual* (Chang et al., 2007). We now add an explanation on lines 140-141.

*4. Line 167-169, Is it correct that the way the authors describe the citation.*

**Response:** OK, we wrote this sentence (see Lines 169-171).

*5. Fig.7 I do not think this figure is necessity.*

**Response:** OK, we removed this figure in the new version, and rearrange the number of the rest figures.

Thanks again for editor's and reviewer's effort on our manuscript!

Best regards,

Zongxing Zou, Huiming Tang, Robert E. Criss, Xinli Hu, Chengren Xiong, Qiong Wu, Yi Yuan

October 5, 2020